# Video Pretraining Advances 3D Deep Learning on Chest CT Tasks

**Alexander Ke**[1]                                                    ALEXKE@CS.STANFORD.EDU
[1] *Department of Computer Science, Stanford University*

**Shih-Cheng Huang**[2]                                               MSCHUANG@STANFORD.EDU
[2] *Department of Biomedical Data Science, Stanford University*

**Chloe O'Connell**[3]                                                COCONNELL@MGH.HARVARD.EDU
[3] *Department of Anesthesia, Massachusetts General Hospital*

**Michał Klimont**[4]                                                 MICHAL.KLIMONT@GMAIL.COM
[4] *Center for Artificial Intelligence in Medicine & Imaging, Stanford University*

**Serena Yeung**[1,2,4,5]                                             SYYEUNG@STANFORD.EDU
[5] *Department of Electrical Engineering, Stanford University*

**Pranav Rajpurkar**[5]                                               PRANAV_RAJPURKAR@HMS.HARVARD.EDU
[5] *Department of Biomedical Informatics, Harvard Medical School*

**Editors:** Accepted for publication at MIDL 2023

## Abstract

Pretraining on large natural image classification datasets such as ImageNet has aided model development on data-scarce 2D medical tasks. 3D medical tasks often have much less data than 2D medical tasks, prompting practitioners to rely on pretrained 2D models to featurize slices. However, these 2D models have been surpassed by 3D models on 3D computer vision benchmarks since they do not natively leverage cross-sectional or temporal information. In this study, we explore whether natural video pretraining for 3D models can enable higher performance on smaller datasets for 3D medical tasks. We demonstrate video pretraining improves the average performance of seven 3D models on two chest CT datasets, regardless of finetuning dataset size, and that video pretraining allows 3D models to outperform 2D baselines. Lastly, we observe that pretraining on the large-scale out-of-domain Kinetics dataset improves performance more than pretraining on a typically-sized in-domain CT dataset. Our results show consistent benefits of video pretraining across a wide array of architectures, tasks, and training dataset sizes, supporting a shift from small-scale in-domain pretraining to large-scale out-of-domain pretraining for 3D medical tasks. Our code is available at: https://github.com/rajpurkarlab/chest-ct-pretraining

## 1. Introduction

Computed tomography (CT) imaging has transformed clinical decision-making, with over 80 million scans performed in the US annually and growing at 12% year over year (Smith-Bindman et al., 2019; Brenner and Hricak, 2010). To keep pace with this expansion, deep learning can assist clinicians with CT interpretation and has translated to FDA approvals, such as Viz.ai's stroke detection from head CTs (Matsoukas et al., 2022). Large labeled datasets are critical to these successes, but curating them is time- and cost-intensive: even the largest medical imaging datasets are much smaller than natural image datasets.

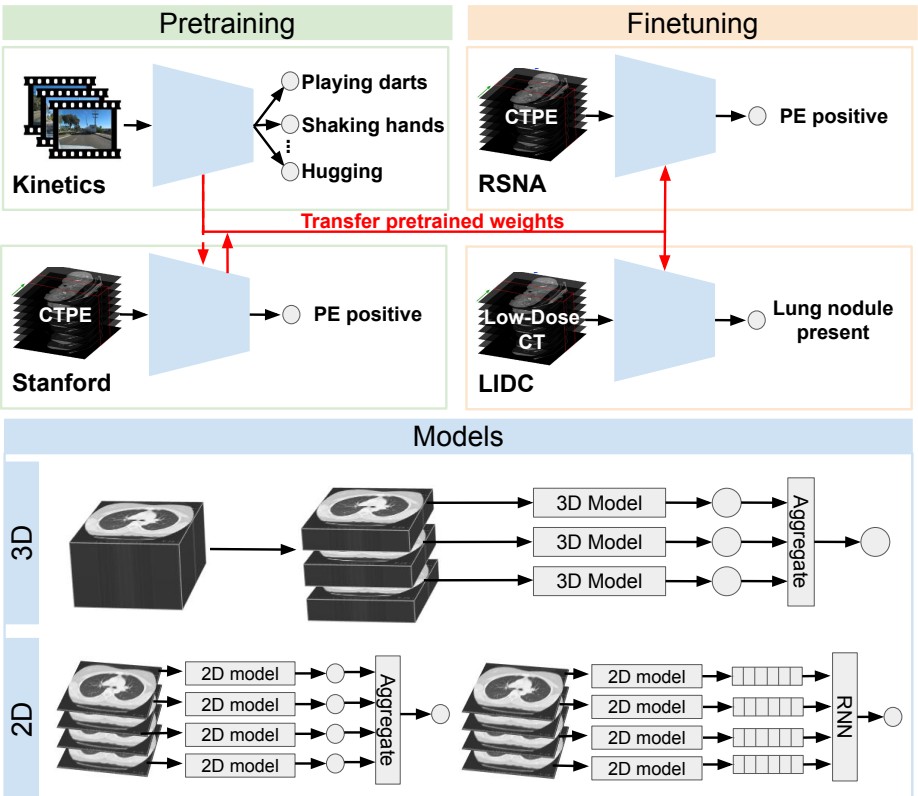

Figure 1: Visual summary of our methods. We examined seven 3D (three 2D) models that were pre-trained either on Kinetics (ImageNet), or on the Stanford dataset for PE detection, or on both sequentially. We then transferred these weights and finetuned them on RSNA for PE detection and LIDC for lung nodule detection.

While 2D medical imaging tasks use ImageNet pretraining *de-facto* (Ke et al., 2021; Raghu et al., 2019), 3D medical imaging tasks have yet to find such a standard. One popular pretraining approach adapts 2D or 2.5D models to 3D tasks (Tajbakhsh et al., 2015; Roth et al., 2016). While these approaches can leverage pretraining on large image datasets like ImageNet, they cannot natively incorporate cross-sectional or temporal information. Furthermore, Yang et al. (2021)'s approach transferring pretrained weights from 2D to 3D CNNs is relatively inflexible in supporting newer architectures (e.g. transformers).

One approach to natively learn cross-sectional information with 3D models is by pretraining on natural videos before finetuning on medical tasks. A few isolated studies have hinted at the utility of large-scale video pretraining. For example, C3D on sports videos for HPV status prediction (Lang et al., 2021), or natural video (Kinetics) for appendicitis and pulmonary embolism (PE) (Rajpurkar et al., 2020; Huang et al., 2020). However, these existing studies focused on performance on a specific clinical task, and it remains unknown whether the utility of large-scale video pretraining is universal across models and tasks.

Beyond video pretraining, pretraining on in-domain CT datasets can capture domain-specific 3D relationships but is typically limited by labeled data availability. Nevertheless,

in-domain pretraining has shown promise for some CT tasks, including PE detection, nodule detection, and liver segmentation (Chen et al., 2019; Gibson et al., 2018; Huang et al., 2020). However, like video pretraining, studies proposing in-domain CT pretraining have largely focused on training a single 3D model for a single clinical task, and do not evaluate the benefits of in-domain pretraining across models and tasks.

**Contributions** We study the impact of (1) video pretraining, in-domain pretraining, and sequential pretraining for a (2) broad, representative universe of models (seven 3D and three 2D) on (3) two large-scale public datasets of chest CTs for PE detection and lung nodule detection, across (4) three dataset sizes (Figure 1). Our study advances this area through four major contributions to training methodology:

(1) While prior work focus on one model for one task, we study a diverse and representative set of models with hyperparameter search, helping generalizing across models;

(2) Further, we are the first to contextualize these findings across classification tasks and protocols, helping generalizing to the chest CT anatomy;

(3) Our direct and original comparisons of video with in-domain and sequential pretraining help disentangle the effect of video pretraining from other pretraining procedures; and

(4) Our experiments illuminate how pretraining's benefits scale with dataset size and how pre- and post- training methods interact with performance, which are especially important in the small data regimes of medicine.

## 2. Methods

**Finetuning datasets** We study PE detection using the RSNA PE CT dataset (Colak et al., 2021), splitting the publicly labeled 7,279 studies (one study per patient) into 5,095 studies for training, 1,092 studies for validation, and 1,092 studies for testing. We further validate our results on lung nodule detection using the LIDC-IDRI dataset (Armato et al., 2011), splitting 1,018 studies (1,010 patients) into 714 studies (707 patients) for training, 152 studies (151 patients) for validation, and 152 studies (152 patients) for testing, with no patient overlap. To understand whether pretraining benefits smaller datasets more, we evaluated with 100% (5,095 studies), 10% (509 studies), and 1% (50 studies) of the RSNA training set and 100% (714 studies) and 10% (71 studies) of the LIDC training set. Each RSNA or LIDC study contains both study-level and slice-level annotations for the presence or absence of PE. Slice-level annotations are used to supervise our models that inference on the slice or window level, while study-level annotations are used to compute performance metrics after aggregating (using the max function) into window-level predictions for a study.

**Pretraining datasets** We pretrain on in-domain, out-of-domain, and sequential out-of-domain then in-domain datasets. For in-domain, we use CT scans from the RadFusion dataset, containing 1,837 studies from Stanford Medicine (Zhou et al., 2021a). We removed the CTs that overlapped with RSNA, leaving 1,241 studies (449 PE positive, 792 PE negative), which we split into training (868 studies), validation (186 studies) and test (187 studies) sets. For out-of-domain, we use ImageNet for our 2D models, and Kinetics-400, a dataset of ∼ 10-second YouTube clips for action recognition for our 3D models, using public weights from their respective authors or PyTorchVideo (Fan et al., 2021a).

**Model universe** We chose 3D models released after 2018 with target $\sim$ 30M parameters from PyTorchVideo's model zoo, resulting in five models: MViT (Fan et al., 2021b), R(2+1)D R50 (Tran et al., 2017), CSN R101 (Tran et al., 2019), SlowFast R50, Slow R50 (Feichtenhofer et al., 2018). We also add Swin-T (Liu et al., 2021) noting recent progress by Swin Transformers, and PENet (Huang et al., 2020) as an architecture developed for PE detection, totalling seven models in our 3D model universe. We benchmark against three 2D models, with ImageNet weights available on torchvision (Paszke et al.): ResNet-18, ResNext-101, and LRCN (an ResNext-101 backbone connected through GRU layers). See Appendix A for detailed description of datasets, training, and evaluation procedure.

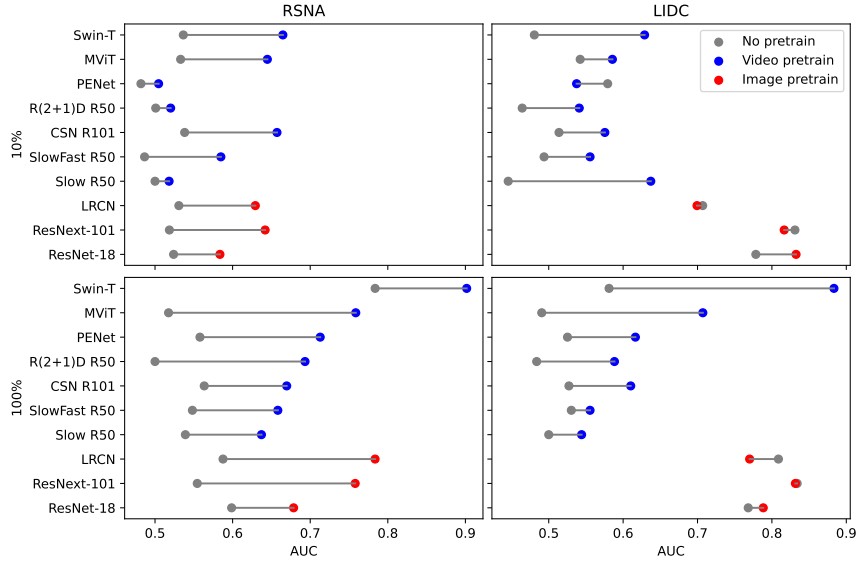

Figure 2: Video pretraining consistently improves 3D models' AUC on 10% and 100% of RSNA and LIDC, except MViT on 10% of LIDC ($p = 0.254$)

## 3. Experiments

### 3.1. Video pretraining improves performance

We first studied whether video pretraining improves the performance of 3D models by testing the downstream performance of 10 models on RSNA and LIDC (Figure 2 and Table 1). On the PE detection task (RSNA), video pretraining improved the mean area under the receiver operating characteristic curve (AUC) of 3D models finetuned on 100% of the training set by 0.146 (95% CI [0.098, 0.241]), from an average AUC of 0.573 [0.500, 0.784] without video pretraining to an average AUC of 0.719 [0.637, 0.902] with video pretraining. Every model benefitted from video pretraining, but these individual differences were not assessed for statistical significance. For models finetuned on 10% of the training set, we observed similar mean improvement of 0.074 [0.018, 0.128] AUC, smaller than the mean 0.146 AUC improvement from finetuning on 100% of the training set.

We also saw performance increases on the nodule detection task (LIDC), on which video pretraining improved the mean AUC of models finetuned on 100% of the training set by 0.124 [0.025, 0.302]. For models finetuned on 10% of the training set, video pretraining had a mean 0.077 [-0.042, 0.192] AUC increase, again smaller than the improvement observed on 100% finetuning.

**Sequential video then CT pretraining outperforms just CT pretraining** We conducted a sensitivity analysis to see whether video pretraining still improves downstream performance when models are further pretrained on the in-domain Stanford CT dataset, comparing 3D model performance with sequential video then pretraining against performance with just CT pretraining (Table 1).

On the nodule detection task with 100% finetuning, models with video then CT pretraining had mean AUC 0.146 [-0.003, 0.358] greater than models with just CT pretraining. We found a difference of 0.055 [-0.031, 0.155] on nodule detection with 10% finetuning. Similarly on the PE detection task, video then CT pretraining improved the mean AUC of 3D models by 0.069 [-0.194, 0.258] with 100% finetuning and 0.002 [-0.071, 0.071] with 10% finetuning, beyond just CT pretraining.

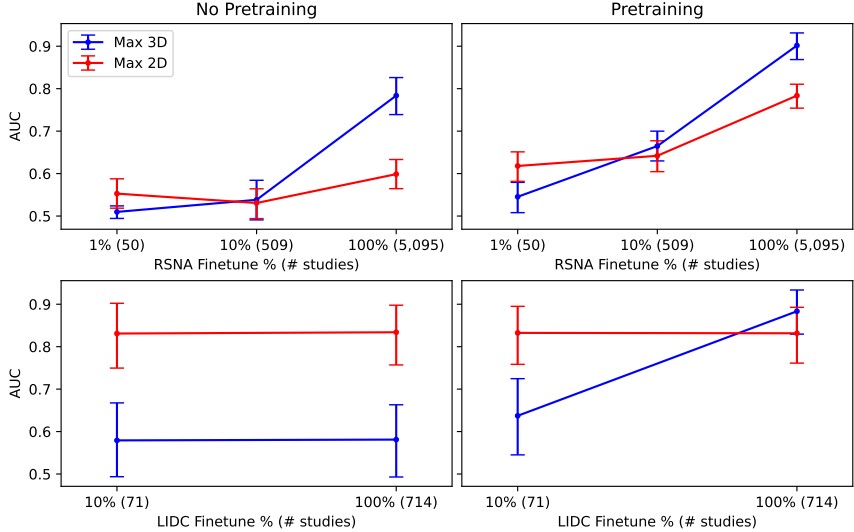

Figure 3: With pretraining, the maximum 3D model AUC is greater than the maximum 2D model AUC on less data compared to without pretraining. Error bars show 95% confidence intervals, and wider confidence intervals for LIDC are due to its relatively smaller test set size (7.3x less data than RSNA).

### 3.2. Video pretraining allows 3D models to outperform 2D baselines

Since we've observed the effect of pretraining varies between 2D and 3D models and with dataset size, we compare the maximum performance of our 3D model universe and our 2D model universe finetuning with 100%, 10%, and 1% on the PE detection task, and with 100% and 10% on the nodule detection task (Figure 3).

With pretraining (video pretraining for 3D models and ImageNet for 2D), we found that the maximum 3D model AUC exceeds the maximum 2D model AUC on 10% of RSNA with difference 0.023 [-0.017, 0.067] AUC, on 100% of RSNA with difference 0.118 [0.079, 0.155] AUC, and on 100% of LIDC with difference 0.052 [-0.006, 0.112] AUC. However, without pretraining maximum 3D model AUC exceeded the maximum 2D model AUC on 10% of RSNA with smaller difference 0.007 [-0.050, 0.060] AUC and on 100% of RSNA with difference 0.185 [0.132, 0.239]. Without pretraining, 3D model performance did not exceed 2D model performance on LIDC.

With pretraining, multiple 3D models exceed the best performing 2D model for a given task and finetuning proportion. The best performing 3D model (Swin-T) outperformed the best performing 2D model (LRCN) by 0.118 [0.079, 0.155] AUC on the PE detection task with finetuning at 100%. With 10% finetuning, the top three 3D models performed better than the best performing 2D model (ResNext-101): Swin-T with difference 0.023 [-0.017, 0.067] AUC, CSN R101 with difference 0.015 [-0.026, 0.060] AUC, and MViT with difference 0.003 [-0.039, 0.046] AUC. On nodule detection with 100% finetuning, the best performing 3D model (Swin-T) also outperformed the best-performing 2D model (ResNext-101) by 0.052 [-0.006, 0.112] AUC.

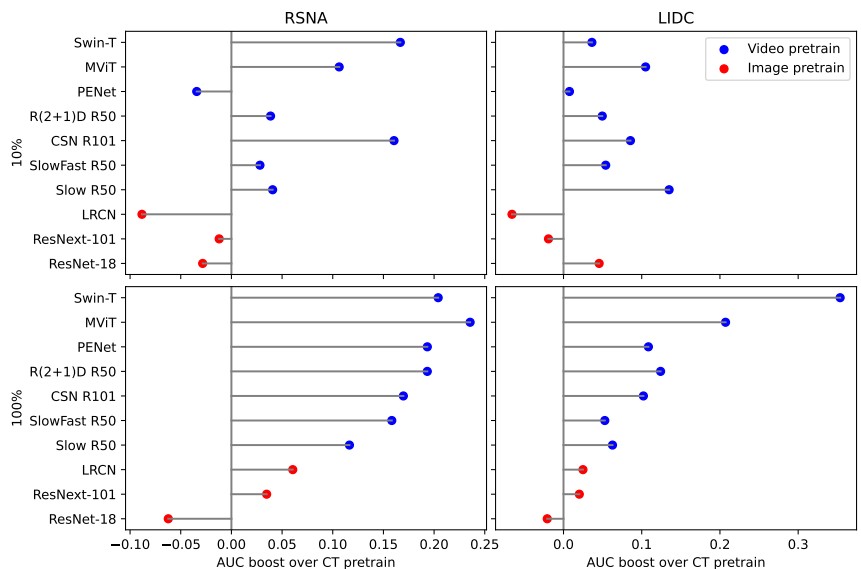

Figure 4: Difference between video/image pretrained AUC and CT pretrained AUC on 10% and 100% of RSNA and LIDC. All 3D models except PENet on 10% of RSNA benefit from video pretraining more than CT pretraining.

### 3.3. Video pretraining outperforms small-scale CT pretraining

Finally, we investigated whether video pretraining on the large-scale Kinetics-400 could improve the performance of 3D models more than pretraining on the comparatively smaller-scale Stanford CT dataset. This CT dataset is in-domain for the RSNA and LIDC datasets,

and is the exact same task as RSNA. We evaluated our models on 100% and 10% finetuning from the RSNA and LIDC training sets (Figure 4 and Table 1).

Comparing video pretraining against CT pretraining, models with video pretraining had mean AUC 0.182 [0.116, 0.236] greater than models with CT pretraining on PE detection with 100% finetuning. We found similar differences of 0.072 [-0.034, 0.167] on PE detection with 10% finetuning, 0.144 [0.053, 0.354] on nodule detection with 100% finetuning, and 0.068 [0.008, 0.135] on nodule detection with 10% finetuning.

Although video pretraining outperforms CT pretraining in isolation, we further investigated whether CT pretraining may be a useful additional step beyond video pretraining. We found that video pretraining outperforms video then CT pretraining by 0.112 [-0.023, 0.398] mean AUC on PE detection with 100% finetuning, 0.070 [-0.034, 0.165] on PE detection with 10% finetuning, -0.002 [-0.149, 0.075] on nodule detection with 100% finetuning, and 0.012 [-0.142, 0.075] on nodule detection with 10% finetuning.

In contrast to our results for 3D models, we found that CT pretraining is useful as additional supervision beyond ImageNet pretraining for 2D models. ImageNet then CT pretraining outperformed (but not statistically significantly) just ImageNet pretraining by 0.025 [-0.014, 0.095] mean AUC on PE detection with 100% finetuning and 0.066 [0.000, 0.105] mean AUC on PE detection with 10% finetuning. We saw smaller boosts on nodule detection: 0.003 [-0.004, 0.013] mean AUC for 100% finetuning, and 0.010 [-0.079, 0.079] mean AUC for 10% finetuning.

## 4. Discussion

### 4.1. Video pretraining improves performance

We demonstrated that natural video pretraining significantly improves performance across multiple models and multiple chest CT tasks. Pretraining on large-scale natural video datasets such as Kinetics have supported progress on human action recognition benchmarks (Carreira and Zisserman, 2017). However, previous studies of out-of-domain pretraining for CT tasks have only leveraged pretraining on ImageNet (Chaunzwa et al., 2021; Parakh et al., 2019; He et al., 2020), pretraining on much smaller action recognition datasets for a focal model (Zunair et al., 2021), or pretraining on Kinetics for a focal model (Rajpurkar et al., 2020; Huang et al., 2020).

Similar to how 2D models benefit from ImageNet pretraining (Ke et al., 2021), this improvement may be rooted in the ability of 3D models to learn spatiotemporal features (Qian et al., 2021) and transfer understanding of temporal relationships from video to spatial relationships. For example, Li et al. (2021) showed that third-person video has latent signals relevant to the first-person, and thus more spatial, perspective. Additionally, radiologists often interpret 3D imaging studies by scrolling through contiguous 2D slices to make abnormalities more apparent.

We further explored the effect of training data size on video pretraining performance gains, an experiment identified but not executed by prior work (Rajpurkar et al., 2020). Surprisingly, we found that video pretraining is more effective on larger datasets for both PE and nodule detection, which differs 2D medical tasks where more training data makes pretraining less effective (Irvin et al., 2019). This could be because 3D medical imaging datasets are an order of magnitude smaller than 2D, so overfitting on smaller datasets

undermines the benefits of pretraining. For example, CheXpert consists of 224,316 chest x-rays, while DeepLesion consists of only 10,594 CT scans (Yan et al., 2018).

## 4.2. Video pretraining allows 3D models to outperform 2D baselines

We found that with video pretraining, 3D models are able to outperform our 2D baselines on smaller datasets and across multiple chest CT tasks. Practitioners often favor 2D models because of their greater performance on smaller datasets (Ebrahimi et al., 2020; Gao et al., 2017), even though large-scale supervised or self-supervised pretraining for a specific 3D model has outperformed some 2D models (Zhou et al., 2021b; Lang et al., 2021). From the recent RSNA PE competition, all of the top 10 submissions used 2D models, with some reporting that 2D models outperformed the 3D models they studied. Only the 8th and 10th best submissions incorporated a 3D model (without video pretraining) (noa).

Currently, modeling for CT tasks often uses 2D models to featurize each slice, which ignores cross-sectional information unlike 3D models. In fact, we selected our 2D model universe (ResNet, ResNext, and LRCN) based on their success on 3D medical tasks (Rajpurkar et al., 2020; Ebrahimi et al., 2020; noa). 3D architectures have not been recently popular for classification and few have investigated their pretraining (Domingues et al., 2020).

In the human action recognition domain, 2D models applied to individual frames were solid performers when datasets were small (Tran et al., 2018), but with larger datasets, 3D models consistently outperform 2D feature extractors (Carreira and Zisserman, 2017; Hara et al., 2018). We speculate that a similar shift from 2D to 3D models may be in store for 3D medical tasks, and we demonstrate that supervision from a large dataset of natural videos yields significant performance boosts that make 3D models immediately competitive with 2D models on the same dataset sizes.

## 4.3. Video pretraining outperforms small-scale CT pretraining

We observed that natural video pretraining yields greater downstream performance than supervised pretraining on a typically-sized dataset of chest CTs. Because 3D models require much larger datasets to train (Carreira and Zisserman, 2017), Kinetics-400 was found to be the first dataset of sufficient scale to avoid overfitting for 3D models in action recognition (Hara et al., 2018). Thus despite the domain gap, natural videos are one of the few datasets with sufficient scale to tame 3D models for 3D medical tasks.

Further, self-supervised learning (SSL) has shown promise in addressing labeled data scarcity. On 2D medical tasks, SSL has been suggested as a complementary or subsequent technique to supervised pretraining (Azizi et al., 2021, 2022). Islam et al. (2021) evaluates SSL against ImageNet pretraining on slices from the RSNA PE detection task, but does not reason about 3D models or video pretraining. On 3D medical tasks, Zhou et al. (2021b) demonstrate outperformance of their in-domain SSL method against supervised 2D and 3D approaches, but because the self vs. fully supervised decision can interact with the in- vs. out-of-domain data decision, they are not equipped for supervised comparisons. Although we constrain our conclusions to supervised pretraining, we study it thoroughly and suggest future work may investigate the interactions and compositionality between SSL and our in-domain and out-of-domain pretraining strategies.

In contrast to our findings for 3D models, CT pretraining provided useful supervision beyond that from natural image classification for our 2D models. Our observed boosts were larger on the PE detection task than the nodule detection task, as expected since our CT pretraining task was PE detection.

## 5. Conclusion

Video pretraining has several practical clinical implications. First, large public video datasets do not contain sensitive patient information, thus comparably large CT datasets may not be public. Video supervision thus could aid development of more effective CT imaging models. Second, video pretraining is appropriate for 3D medical imaging tasks where data is often scarce, such as new or rare conditions. The success of video pretraining for CT suggests investigation into its benefits for ultrasound, MRI, and other 3D modalities.

In conclusion, we demonstrate the utility of video pretraining through, to the best of our knowledge, the first broad-scale study across 3D models, chest CT tasks, pretraining setups, and dataset sizes. We verify that only datasets of sufficient scale as Kinetics-400, even though its task is out-of-domain, can unlock 3D models' performance to outperform 2D baselines. We expect that this work will encourage the more widespread use of video pretraining to enable improvements in 3D medical imaging tasks using 3D models.

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

# Appendix A. Detailed Methods

## A.1. Data

### A.1.1. PRETRAINING DATASETS

Our 3D models are pretrained on Kinetics-400, a dataset containing 306,245 YouTube clips each around 10 seconds long. Like in 2D transfer learning, we replace the 400-way classification head with a single output head for our downstream tasks regularized with dropout.

We pretrain on CT scans from the in-domain RadFusion dataset (henceforth, the Stanford dataset), containing 1,837 1.25 mm axial CT studies from Stanford Medicine (Zhou et al., 2021a). 1,241 studies (449 PE positive, 792 PE negative) remained after removing CT scans that overlapped with the RSNA dataset. We randomly split patients into training (868 studies), validation (186 studies) and test (187 studies) sets.

### A.1.2. Finetuning datasets

We study PE detection using the RSNA PE CT dataset, the largest publicly available PE dataset sourced across five international medical centers and annotated by 80 subspecialist thoracic radiologists (Colak et al., 2021). Among the publicly labeled 7,279 studies, with one study per patient, the dataset was split into 5,095 studies for training, 1,092 studies for validation, and 1,092 studies for testing. We use axial series with a slice thickness ranging from 0.625 to 5.0 mm. Each study contains both study-level and slice-level annotations for the presence or absence of PE. Slice-level annotations are used to supervise our models that inference on the slice or window level, while study-level annotations are used to compute performance metrics after aggregating window-level predictions for a study

We further validate our results on lung nodule detection using the LIDC-IDRI dataset, a large publicly available dataset sourced from seven medical centers and eight medical imaging companies and annotated by 12 thoracic radiologists (Armato et al., 2011). Among the 1,018 studies from 1,010 patients, the dataset was split into 714 studies from 707 patients for training, 152 studies from 151 patients for validation, and 152 studies from 152 patients for testing, with no patient overlap between splits. We use the axial series with slice thickness ranging from 0.6 to 4 mm. Each study contains both study-level and slice-level annotations for nodules $\geq$ 3mm from four radiologists, and we consider a slice as positive for a nodule if at least three radiologists labeled the slice as positive. Similar to the RSNA PE dataset, we use slice-level annotations to supervise our models and study-level annotations for evaluation.

## A.2. Training and Evaluation Procedure

We use public checkpoints of 3D models pretrained on Kinetics-400 available on PyTorchVideo's model zoo (Fan et al., 2021a). We chose 3D models released after 2018 with target $\sim$ 30M parameters, resulting in five models: MViT (Fan et al., 2021b), R(2+1)D R50 (Tran et al., 2017), CSN R101 (Tran et al., 2019), SlowFast R50, Slow R50 (Feichtenhofer et al., 2018). Noting the recent benchmark progress made by Swin Transformers, we also study Swin-T with official Kinetics-400 weights (Liu et al., 2021). Finally, we incorporate an architecture developed for PE detection (PENet) with official Kinetics-400 weights (Huang et al., 2020). In total, this yields seven models in our 3D model universe. We benchmark these performances against three 2D models, with ImageNet weights available on torchvision (Paszke et al.): ResNet-18, ResNext-101, and LRCN (which uses a ResNext-101 backbone connected through GRU layers).

Before training, slices consisting of raw Hounsfield Units are clipped to the ranges [400, 1000] for PE detection and [-600, 1500] for nodule detection and zero-centered. During training, we resize each 512 x 512 pixel slice to 256 x 256 for computational efficiency, apply an AutoAugment policy learned on ImageNet (Cubuk et al., 2019), randomly rotate up to 20 degrees, and randomly crop to 224 x 224. We upsample positive studies to have equal prevalence as negative studies during training. Instead of using the entire volumetric CT scan, we use a sliding window of 32 consecutive slices as inputs to our 3D models (24 for PENet, as recommended by the authors (Huang et al., 2020)). A sliding window is considered positive if at least four slices are labeled as positive.

We optimize the binary cross-entropy loss using Adam ($\beta_1 = 0.9$, $\beta_2 = 0.999$). We use an exponentially sampled grid search of learning rates over $10^{-1}$ to $10^{-5}$, with a learning rate schedule that halves the learning rate on validation loss plateau for 5 epochs. We choose the learning rate that achieves the highest validation accuracy on 10% of a dataset for computational efficiency and use that learning rate for 100% of the same dataset. We use a batch size of 32 for 2D models and 16 for 3D models. We use half precision (16-bit precision) to train our models in order to lower memory requirements.

Slice- and window-level predictions are aggregated to study-level predictions using the max function (if any window has a positive prediction, then the study has a positive prediction). We train our models for a maximum of 50 epochs, with early stopping if study-level validation AUC does not improve for five epochs. The model that achieves the greatest validation AUC is chosen for evaluation on the test set.

We use the nonparametric bootstrap to estimate 95% confidence intervals for each statistic. For AUC as an example, we draw 1,000 replicates with replacement from the test set and calculate the AUC on each replicate. We report the 2.5 and 97.5 percentiles of the resulting distribution as our confidence interval. Statistical significance is assessed at the 0.05 level.

# Appendix B. AUC [95% CI] for model universe on all training setups

Table 1: The Video/Image column denotes if out-of-domain pretraining was used: video for 3D models and image for for 2D models. The CT column denotes if CT pretraining was used.

| Video /Image | CT | Down -stream | Fine -tune | ResNet-18 | ResNext-101 | LRCN | PENet |
|---|---|---|---|---|---|---|---|
| yes | yes | RSNA | 10% | 0.689 [0.654, 0.723] | 0.642 [0.605, 0.677] | 0.722 [0.691, 0.754] | 0.539 [0.500, 0.576] |
| | | | 100% | 0.774 [0.743, 0.803] | 0.753 [0.719, 0.782] | 0.770 [0.738, 0.797] | 0.527 [0.482, 0.569] |
| | | LIDC | 10% | 0.754 [0.665, 0.832] | 0.847 [0.781, 0.909] | 0.778 [0.698, 0.848] | 0.679 [0.590, 0.765] |
| | | | 100% | 0.801 [0.727, 0.868] | 0.833 [0.760, 0.898] | 0.766 [0.685, 0.840] | 0.654 [0.562, 0.740] |
| | no | RSNA | 1% | 0.543 [0.506, 0.578] | 0.544 [0.508, 0.579] | 0.618 [0.582, 0.651] | 0.476 [0.441, 0.514] |
| | | | 10% | 0.584 [0.546, 0.620] | 0.642 [0.605, 0.677] | 0.629 [0.592, 0.665] | 0.505 [0.484, 0.524] |
| | | | 100% | 0.679 [0.644, 0.713] | 0.758 [0.723, 0.789] | 0.784 [0.754, 0.810] | 0.713 [0.694, 0.733] |
| | | LIDC | 10% | 0.832 [0.758, 0.895] | 0.817 [0.735, 0.891] | 0.699 [0.603, 0.786] | 0.537 [0.450, 0.628] |
| | | | 100% | 0.788 [0.711, 0.859] | 0.832 [0.761, 0.893] | 0.770 [0.685, 0.848] | 0.617 [0.525, 0.706] |
| no | yes | RSNA | 10% | 0.612 [0.578, 0.648] | 0.654 [0.618, 0.690] | 0.718 [0.684, 0.748] | 0.539 [0.500, 0.576] |
| | | | 100% | 0.741 [0.706, 0.773] | 0.723 [0.691, 0.755] | 0.723 [0.690, 0.754] | 0.520 [0.482, 0.556] |
| | | LIDC | 10% | 0.787 [0.707, 0.854] | 0.836 [0.763, 0.899] | 0.765 [0.677, 0.842] | 0.530 [0.440, 0.619] |
| | | | 100% | 0.809 [0.729, 0.882] | 0.811 [0.732, 0.884] | 0.745 [0.658, 0.823] | 0.508 [0.412, 0.606] |
| | no | RSNA | 1% | 0.492 [0.462, 0.521] | 0.492 [0.462, 0.521] | 0.553 [0.519, 0.588] | 0.476 [0.441, 0.513] |
| | | | 10% | 0.524 [0.487, 0.558] | 0.519 [0.483, 0.554] | 0.531 [0.494, 0.564] | 0.482 [0.461, 0.503] |
| | | | 100% | 0.599 [0.564, 0.633] | 0.554 [0.520, 0.590] | 0.588 [0.554, 0.622] | 0.558 [0.514, 0.604] |
| | | LIDC | 10% | 0.778 [0.693, 0.849] | 0.831 [0.750, 0.902] | 0.707 [0.622, 0.787] | 0.579 [0.494, 0.668] |
| | | | 100% | 0.768 [0.689, 0.839] | 0.834 [0.757, 0.898] | 0.809 [0.728, 0.879] | 0.525 [0.440, 0.615] |

| Video /Image | CT | Down -stream | Fine -tune | MViT | Swin-T | Slow R50 | SlowFast R50 |
|---|---|---|---|---|---|---|---|
| yes | yes | RSNA | 10% | 0.609 [0.571, 0.643] | 0.500 [0.500, 0.500] | 0.479 [0.441, 0.514] | 0.486 [0.449, 0.523] |
| | | | 100% | 0.781 [0.643, 0.810] | 0.504 [0.468, 0.542] | 0.500 [0.500, 0.500] | 0.615 [0.577, 0.653] |
| | | LIDC | 10% | 0.532 [0.445, 0.625] | 0.561 [0.470, 0.649] | 0.562 [0.467, 0.656] | 0.488 [0.429, 0.551] |
| | | | 100% | 0.702 [0.620, 0.780] | 0.888 [0.833, 0.937] | 0.500 [0.500, 0.500] | 0.500 [0.500, 0.500] |
| | no | RSNA | 1% | 0.516 [0.480, 0.554] | 0.517 [0.479, 0.555] | 0.545 [0.508, 0.579] | 0.482 [0.446, 0.518] |
| | | | 10% | 0.645 [0.603, 0.678] | 0.665 [0.630, 0.700] | 0.518 [0.468, 0.563] | 0.585 [0.537, 0.637] |
| | | | 100% | 0.759 [0.725, 0.792] | 0.902 [0.869, 0.931] | 0.637 [0.587, 0.685] | 0.658 [0.605, 0.706] |
| | | LIDC | 10% | 0.586 [0.492, 0.682] | 0.629 [0.537, 0.718] | 0.637 [0.545, 0.725] | 0.556 [0.458, 0.651] |
| | | | 100% | 0.707 [0.616, 0.791] | 0.883 [0.830, 0.934] | 0.544 [0.452, 0.645] | 0.556 [0.464, 0.655] |
| no | yes | RSNA | 10% | 0.538 [0.501, 0.572] | 0.498 [0.461, 0.534] | 0.477 [0.450, 0.504] | 0.557 [0.522, 0.593] |
| | | | 100% | 0.523 [0.572, 0.560] | 0.698 [0.662, 0.730] | 0.521 [0.495, 0.546] | 0.500 [0.500, 0.500] |
| | | LIDC | 10% | 0.481 [0.438, 0.519] | 0.593 [0.510, 0.674] | 0.502 [0.423, 0.583] | 0.502 [0.416, 0.582] |
| | | | 100% | 0.500 [0.500, 0.500] | 0.530 [0.446, 0.610] | 0.482 [0.403, 0.558] | 0.503 [0.404, 0.595] |
| | no | RSNA | 1% | 0.510 [0.494, 0.524] | 0.499 [0.462, 0.539] | 0.500 [0.500, 0.500] | 0.499 [0.495, 0.500] |
| | | | 10% | 0.533 [0.495, 0.569] | 0.536 [0.488, 0.581] | 0.500 [0.500, 0.500] | 0.487 [0.442, 0.525] |
| | | | 100% | 0.517 [0.481, 0.557] | 0.784 [0.739, 0.826] | 0.539 [0.493, 0.586] | 0.548 [0.505, 0.597] |
| | | LIDC | 10% | 0.542 [0.447, 0.640] | 0.481 [0.422, 0.540] | 0.446 [0.362, 0.528] | 0.494 [0.407, 0.586] |
| | | | 100% | 0.491 [0.400, 0.584] | 0.581 [0.493, 0.663] | 0.500 [0.500, 0.500] | 0.530 [0.437, 0.629] |

| Video /Image | CT | Down -stream | Fine -tune | CSN R101 | R(2+1)D R50 | | |
|---|---|---|---|---|---|---|---|
| yes | yes | RSNA | 10% | 0.510 [0.475, 0.545] | 0.479 [0.443, 0.516] | | |
| | | | 100% | 0.677 [0.645, 0.713] | 0.641 [0.605, 0.673] | | |
| | | LIDC | 10% | 0.645 [0.550, 0.727] | 0.507 [0.408, 0.600] | | |
| | | | 100% | 0.535 [0.450, 0.627] | 0.738 [0.661, 0.816] | | |
| | no | RSNA | 1% | 0.502 [0.466, 0.538] | 0.487 [0.449, 0.525] | | |
| | | | 10% | 0.657 [0.620, 0.690] | 0.520 [0.475, 0.565] | | |
| | | | 100% | 0.670 [0.618, 0.719] | 0.693 [0.649, 0.739] | | |
| | | LIDC | 10% | 0.575 [0.515, 0.637] | 0.541 [0.483, 0.600] | | |
| | | | 100% | 0.610 [0.518, 0.697] | 0.588 [0.490, 0.678] | | |
| no | yes | RSNA | 10% | 0.497 [0.492, 0.501] | 0.482 [0.466, 0.498] | | |
| | | | 100% | 0.500 [0.500, 0.500] | 0.500 [0.500, 0.500] | | |
| | | LIDC | 10% | 0.490 [0.401, 0.581] | 0.491 [0.397, 0.587] | | |
| | | | 100% | 0.508 [0.415, 0.599] | 0.464 [0.378, 0.564] | | |
| | no | RSNA | 1% | 0.500 [0.500, 0.500] | 0.500 [0.500, 0.500] | | |
| | | | 10% | 0.538 [0.491, 0.584] | 0.501 [0.478, 0.525] | | |
| | | | 100% | 0.563 [0.513, 0.609] | 0.500 [0.500, 0.500] | | |
| | | LIDC | 10% | 0.514 [0.435, 0.597] | 0.464 [0.382, 0.550] | | |
| | | | 100% | 0.527 [0.433, 0.620] | 0.484 [0.388, 0.583] | | |

