# OpenReview forum: "Video pretraining advances 3D deep learning on chest CT tasks"
_MIDL.io/2023/Conference — MIDL 2023 Poster_

### Official Review · Reviewer_SQG3 · 2023-02-02

**Confidence:** 3
**Preliminary Rating:** 5
**Recommendation:** Oral, Poster

**Summary:**

This study presents a study of in-domain and out-of-domain pre-training across the RSNA and LIDC tasks targeting CT image processing. Of interest, they find that pretraining models on 10 second video clips from YouTube  substantially approves performance. Direct comparison with network designs is limited, but the results are intriguing and will likely prompt discussion.

**Strengths:**

+ The paper presents a clever concept for generating new data.
+ The rationale behind using video as an analog for visual search is well reasoned.
+ The authors explore distinct tasks and a range of network designs.

**Weaknesses:**

- The technical details of how the pre training was performed are not presented in detail. Release of source code mitigates, but does not receive this burden.
- A more nuanced discussion of improvements and losses need to be discussed.
- A direct comparison of different approaches of pre-training on the same network would improve clarity.

**Deanonymize Review:**

no

**Paper Type:**

methodological development

**Questions To Address In The Rebuttal:**

Additional technical details on the pre-training process are needed.

A discussion on the absolute performance of the various models (e.g., Fig. 2) along with the relative changes with pre-training is needed.

---

### Official Review · Reviewer_tcL9 · 2023-02-06

**Confidence:** 5
**Preliminary Rating:** 1

**Summary:**

This manuscripts looked into whether natural video pretraining could be beneficial for 3D medical imaging tasks. The authors conducted a series of experiments on two chest CT tasks: PE detection using the RSNA PE CT dataset and lung nodule detection using the LIDC-IDRI dataset. According to their findings, they claimed pretraining models on natural videos can outperform pretrained models on a typical-sized in-domain CT dataset. As a result, they proposed that a paradigm shift from small-scale in-domain pretraining to large-scale out-of-domain pretraining for 3D medical tasks is required.


**Strengths:**

The authors conducted a set of  experiments to assess the effectiveness of natural video pretraining in chest CT tasks, which is good. Their paper's credibility may be enhanced if they pair it with more intelligent and in-depth analyses.


**Weaknesses:**

1) The authors conducted a series of experiments without providing any new insights. The finding that video pretraining is useful for 3D medical imaging has already been demonstrated in many previous studies.

2) The validity and reliability of the findings may be questionable. Previous studies, such as Models Genesis (Zhou et al 2021), have demonstrated that pretraining on 3D medical imaging (in-domain data) outperforms pretraining on natural video data (Kinetics dataset). However, this paper makes the opposite claim without comparing with Models Genesis.

3) The authors ignored self-supervised learning in their experimental settings, which raises serious concerns about the validity of the reported results. As the authors are aware, self-supervised learning is a major branch of research to address the limited availability of annotations in medical imaging.

4) The study fails to address a crucial question regarding the usefulness and superiority of pretraining on natural video data over in-domain data for 3D medical imaging. There is a noticeable difference between the two types of data. Moreover, the reasons for why the authors prioritize supervised training as the pretraining approach are not clear. The potential of self-supervised learning in medical imaging has been demonstrated and it has shown success in utilizing large amounts of unlabeled data for learning generalizable representations.

5) The writing of the paper needs major revisions as it is difficult to follow. The introduction needs to be revised to accurately reflect the contributions of the paper. There are many over-claimed contributions in the paper. The figures and captions also need better captions, as the current captions are too short and vague.

6) Some of the reported results are inconsistent and need clarification. For example, in Figure 2, it is unclear why video pre-training underperforms compared to no-pretraining in the 10% LIDC dataset. Similarly, mixed results are seen for image pretraining in the 100% LIDC dataset.

7) The current experimental setup may not adequately support the authors' claims, as it is limited to PE and lung nodule detection tasks only. The experiments in this study overlook the fact that medical image segmentation tasks are more challenging and have more practical significance compared to classification and detection tasks.

**Deanonymize Review:**

no

**Paper Type:**

validation/application paper

**Questions To Address In The Rebuttal:**

The authors may wish to address the critiques raised in the weaknesses during the rebuttal stage. It is important to note that simply conducting experiments and reporting them without providing new insights may not be considered significant research. To enhance the impact and value of the paper, the authors could consider organizing their experimental setup in a clearer manner and aim to extract new knowledge and insights. In the current deep learning era, the statement that pretraining is useful may not be considered novel, interesting or groundbreaking for the reader.

---

### Official Review · Reviewer_v5KN · 2023-02-06

**Confidence:** 4
**Preliminary Rating:** 4
**Recommendation:** Oral

**Summary:**

This paper explores whether natural video pretraining for 3D models can enable higher performance on smaller datasets for 3D medical tasks.

It should belong to the validation paper.

The conclusion is that pretraining is useful, and a large-size dataset is better than a smaller one.

Validations are conducted on pre-training datasets (Kinetics, Stanford) and fine-tuning datasets (RSNA, LIDC) on two tasks: PE and Lung nodule detection.

**Strengths:**

- A colourful paper with extensive experiments on two different tasks with ten models, including traditional convolutional neural network and Transformers network architecture.
- The conclusion is beneficial to other researchers, which could avoid unnecessary calculation. Like: Video pretraining improves performance; Video pretraining allows 3D models to outperform 2D baselines; Video pretraining outperforms small-scale CT pretraining.


**Weaknesses:**

- The figures utilized to show the AUC difference before and after fine-tuning (Figure 2,4). It is clear to see the improvement after pre-training.
But the exact values are not provided. Does the comparison performed under fair conditions? Is the performance reported once or the average results from several different runs?
- From Figure 3, the std is actually very large. Can authors give more discussion on this? HIgh std sometime means the stability is poor or the model.

**Deanonymize Review:**

no

**Paper Type:**

validation/application paper

**Questions To Address In The Rebuttal:**

I think this paper is interesting.
The main concerns are on the experimental setting and result stability.
If the authors could answer my questions, I would appreciate it.

I would consider increasing my rating according to the response.

---

### Official Review · Reviewer_j6nF · 2023-02-06

**Confidence:** 4
**Preliminary Rating:** 3

**Summary:**

Submission 46 presents an empirical evaluation of various transfer learning scenarios for medical image classification, specifically pertaining to binary classification in common chest CT datasets. It pre-selects a number of 2D slicewise classification models alongside “3D” models (which are taken from the video classification literature) and pretrains them in a supervised manner on large labeled datasets (chest CT and/or natural videos) and finetunes them on target chest CT binary classification. Based on its empirical findings, it makes a number of claims about transfer learning for 3D volume classification.

**Strengths:**

- The proposed scope of the paper is ambitious: it benchmarks seven models, multiple set ups of transfer learning, uses large-scale datasets, etc. It would be useful to people working on chest CT classification if properly executed and presented.
- It’s empirical findings are interesting as they are somewhat unintuitive in that one would not typically expect pretraining on natural videos to yield a strong difference on 3D volume classification.
- Open code is made available for improved reproducibility and analyses are run on public data, which are both crucial in large-scale empirical projects.


**Weaknesses:**

# Major

## A. Unclear motivation
It is highly unintuitive to me as to why: (1) *video* classification networks are being used to classify large-scale 3D volumes and (2) 3D *spatially-aware* networks like simple 3D CNNs or ViTs are completely excluded from the analysis. While, to my knowledge, early work in medical imaging with deep learning and some of the referenced papers use strategies such as (1), I do not understand why properly tuned (see more below) 3D networks are not included if the aim is to show that video networks acting on slices are the best of all worlds.

Further, please correct me if I misunderstood, but it appears that these video and 2D networks were trained only on axial slices. Would something change if training was done on all views and then aggregated? From the appendix, these datasets do not appear to be hugely anisotropic, but I’m happy to be corrected on this.

## B. No mention of established work in this space
There is a large body of literature studying the role of large pretraining datasets in downstream medical image classification. Unfortunately, the submission does not appear to cite, discuss, and/or compare (when appropriate) with the work in this area. For example, [Azizi et al, ICCV21](https://openaccess.thecvf.com/content/ICCV2021/html/Azizi_Big_Self-Supervised_Models_Advance_Medical_Image_Classification_ICCV_2021_paper.html) perform a large-scale empirical analysis of the utility of natural vs medical pretraining corpora and finds the exact opposite trend as this submission. Further, there are dozens of papers on self and semi-supervised pretraining methods used for medical image classification that are not acknowledged in any way in the current version of the paper (recent arbitrary example that can easily be extended to 3D [1](https://openaccess.thecvf.com/content/WACV2023/papers/Xiao_Delving_Into_Masked_Autoencoders_for_Multi-Label_Thorax_Disease_Classification_WACV_2023_paper.pdf)). Please correct me if I’m missing something obvious, but these appear to be large oversights.


## C. Experiments may need more rigor as trends are not clear
As far as I can tell from the appendix, these experiments were run without learning rate decay, multiple runs with multiple random seeds, regularization, and the augmentation policies used were taken from ImageNet (tangentially, how were these augmentations applied to 3D volumes?) and not developed for the binary CT classification task. As a result, the experimental trends presented are ambiguous at best and need to be revisited. For example,
- Fig 2 (bottom left subfigure): How are models such as MViT *getting worse* without pretraining by going from 10% to 100% data availability during finetuning?
- Fig 2 (top right subfigure): Similarly, how is the pretraining/from-scratch gap for SwinT and MViT *widening* with significantly more finetuning data available? If anything, the 10 and 100% data trends should be reversed as less pretraining data is typically required if large finetuning datasets exist.
- Fig 3 (bottom row): How does the red line (max 2d) in the two subfigures (pretraining vs not) have no difference at all?

Moreover, I found it very difficult to get concrete experimental takeaways from this submission. For example, 2D and 3D transfer learning setups have opposite trends (line 1 of page 7). I suggest that future versions of this work be made more rigorous to gain more insight.

## D. Largely unclear presentation
In general, the presentation of the submission can be significantly improved. For example,
- None of the baseline models are actually cited in the text. Outside of obvious and popular models like a resnet, it’s quite unclear what random acronyms for methods such as CSN, R(2+1), etc. are supposed to represent.
- The most important and relevant experimental and implementation details (crucial to a validation project such as this) are moved to the appendix. This leaves only an extremely high-level 3-paragraph summary of the methods in the main paper which leaves the readers unable to discern what is actually being implemented.
- The exposition of the results is, in my opinion, rather narrative and long-winded which makes it hard to read. Further, Section 4 (the discussion) alternates between unnecessary recounts of related work and presenting speculation about its experimental trends as explanation (e.g., the arguments boil down to “a video deep network did well on this one task and that must be because radiologists look at images slicewise”, see section 3.1 of [this paper](https://arxiv.org/pdf/1807.03341.pdf) for why that is problematic).
- Major figure captions are far too short for me to be able to interpret what they are trying to say. For example, Figure 1 (the highlight figure) contains several subfigures without a clear description of how they relate to each other.


**Deanonymize Review:**

no

**Detailed Comments:**

# Minor
- The submission makes several claims with somewhat grandiose phrasing that are not supported by the text and this makes it quite distracting to read. For example, the abstract claims this work to be a “paradigm shift” and the contributions claim that the analyses “illuminate scaling laws” and include a “diverse and representative model universe” (what is a *model universe*?). The writing would be significantly improved if the claims were tempered and restricted to what is actually shown.
- The paper makes references to pretraining. However, as pretraining can also include self-supervised methods, I suggest that it clarifies that it is actually referring to supervised transfer learning.


**Paper Type:**

validation/application paper

**Questions To Address In The Rebuttal:**

If the rebuttal were to correct me on formulational points A & B, and clarify that the concerns in C are not valid as they are addressed somehow without mention in the original submission, I would be happy to revisit my score.

# EDIT AFTER REBUTTAL

I thank the authors for their rebuttal and extensive discussions. I am raising my score to 'borderline'.

My concerns about weaknesses A, B, and D are resolved to a reasonable extent, although I do stress that the title should be modified to include transfer learning or supervised pretraining. If not, the pretraining term is too overloaded with self-supervised learning and just confuses the reader and then necessitates a spate of comparisons with self-supervised work.

However, the response to weakness C (no clear experimental trends) is a bit too speculative for my comfort. I do believe that these experiments need to be run with multiple random seeds to actually assess trends and the rebuttal did not address why the autoaug augmentation policy taken from imagenet was used.

---

### Meta-Review · Area_Chair_yjzb · 2023-02-25

**Recommendation:** Accept (Poster)
**Confidence:** 5

**Metareview:**

This is an empirical work, investigating experimentally whether pre-training neural networks with supervision on natural (non medical) video is beneficial for followup medical image analysis tasks with 2D and 3D models. The work has conducted a significant number of experiments, trying to cover a diverse set of models, dataset sizes, and 2 different medical imaging tasks, in an attempt to observe more generalizable trends. The work finds that such pretraining, on natural vision videos, is benecificial in most settings, which can be interesting in practice as such pretrained models are widely available in practice in model zoos.

The authors have addressed a significant number of concerns from the reviewers. Main remaining cons are about the generalizability of the findings in other tasks, and whether some findings are "solid" given the variance in some results. Regardless, the reviewers seem to agree there is sufficient interesting points in this work.

Overall, I recommend acceptance and believe this experimental work may provide some useful insights to the MIDL attendees and readers.